# Multi-Fold Computational Analysis to Discover Novel Putative Inhibitors of Isethionate Sulfite-Lyase (Isla) from *Bilophila wadsworthia:* Combating Colorectal Cancer and Inflammatory Bowel Diseases

**DOI:** 10.3390/cancers15030901

**Published:** 2023-01-31

**Authors:** Muhammad Waqas, Sobia Ahsan Halim, Atta Ullah, Assim Alaa Mohammed Ali, Asaad Khalid, Ashraf N. Abdalla, Ajmal Khan, Ahmed Al-Harrasi

**Affiliations:** 1Natural and Medical Sciences Research Center, University of Nizwa, Birkat Al Mauz, P.O. Box 33, Nizwa 616, Oman; 2Department of Biotechnology and Genetic Engineering, Hazara University Mansehra, Mansehra 2100, Pakistan; 3Department of Zoology, Abdul Wali Khan University, Mardan 23200, Pakistan; 4Substance Abuse and Toxicology Research Center, Jazan University, P.O. Box 114, Jazan 45142, Saudi Arabia; 5Medicinal and Aromatic Plants and Traditional Medicine Research Institute, National Center for Research, P.O. Box 2404, Khartoum 11111, Sudan; 6Department of Pharmacology and Toxicology, College of Pharmacy, Umm Al-Qura University, P.O. Box 715, Makkah 21955, Saudi Arabia

**Keywords:** glycal radical enzyme, *Bilophila wadsworthia*, keyword, colorectal cancer, colitis

## Abstract

**Simple Summary:**

Hydrogen sulfide (H_2_S) has been produced by certain gut bacteria and associated with the development of inflammatory bowel disease (IBD) and colon cancer. H_2_S is produced by these bacteria regulate the gut inflammation and cell proliferation. The resulted H_2_S produced by an enzyme Isethionate sulfite-lyase (Isla) in the colonial *Bilophila wadsworthia* bacteria. Literature has suggested a potential association between Isla and cancer and the production of hydrogen sulfide (H_2_S). Studies suggest that reducing the H_2_S concentration by targeting Isla is a novel and potential therapeutic target for IBD and colon cancer. We apply structure-based drug-designing approaches for Isla and isolate six compounds from natural products and their synthetic derivatives having improved drug-like properties. The molecular dynamics approach was used to confirm the stability and affinity of the selected compounds. Our finding suggests that these compounds can be possible potential inhibitors for the Isla protein.

**Abstract:**

A glycal radical enzyme called isethionate sulfite-lyase (Isla) breaks the C–S bond in isethionate to produce acetaldehyde and sulfite. This enzyme was found in the Gram-negative, colonial *Bilophila wadsworthia* bacteria. Sulfur dioxide, acetate, and ammonia are produced by the anaerobic respiration route from (sulfonate isethionate). Strong genotoxic H_2_S damages the colon’s mucous lining, which aids in the development of colorectal cancer. H_2_S production also contributes to inflammatory bowel diseases such as colitis. Here, we describe the structure-based drug designing for the Isla using an in-house database of naturally isolated compounds and synthetic derivatives. In structure-based drug discovery, a combination of methods was used, including molecular docking, pharmacokinetics properties evaluation, binding free energy calculations by the molecular mechanics/generalized born surface area (MM/GBSA) method, and protein structure dynamics exploration via molecular dynamic simulations, to retrieve novel and putative inhibitors for the Isla protein. Based on the docking score, six compounds show significant binding interaction with the Isla active site crucial residues and exhibit drug-like features, good absorption, distribution, metabolism, and excretion profile with no toxicity. The binding free energy reveals that these compounds have a strong affinity with the Isla. In addition, the molecular dynamics simulations reveal that these compounds substantially affect the protein structure dynamics. As per our knowledge, this study is the first attempt to discover Isla potential inhibitors. The compounds proposed in the study using a multi-fold computational technique may be verified in vitro as possible inhibitors of Isla and possess the potential for the future development of new medications that target Isla.

## 1. Introduction

Several bacteria present in the human gut can release hydrogen sulfide (H_2_S) as a byproduct of their respiration [1]. The H_2_S significantly affects human health and involves numerous disorders, including inflammatory bowel disease, a weaker colonic mucus barrier, tissue swelling (Crohn’s disease), and colon cancer [2]. In the human body, hydrogen sulfide levels are influenced mainly by gut microbiota [3]. According to recent research, the mucus barrier may be broken by sulfide through a novel method [4]. Certain gut bacteria create sulfur, which potently breaks down disulfide bonds in mucus and lyses the polymeric MUC2 network [4,5]. The mucus layer thus thins and becomes more porous, enabling harmful substances and bacteria in the gut lumen to communicate directly with the surface of epithelial cells and damaging the immune response. According to the available research, both animal and human models of inflammatory bowel disease (IBD) have elevated levels of sulfide-producing bacteria and sulfide itself in the gut, suggesting that altered mucus barrier function may play a role in the development of IBD. Compared to healthy persons, patients with colon cancer have been shown to have a significantly different fecal and colonic microbial composition [6,7,8,9,10,11,12,13,14,15]. High levels of fecal hydrogen sulfide (H_2_S) and bile acids in colon cancer patients, as measured by *Bilophila wadsworthia*, are linked to an overabundance of taurocholate [16]. This microbial gas stimulates colonocyte proliferation and inflammation while blocking the oxidation of the anticarcinogenic metabolite butyrate, a preferred substrate for the colonocytes [17,18,19]. Sulfidogenic bacteria provide an H_2_S-rich environment that promotes colorectal cancer. Therefore, decreasing sulfide levels in the gut lumen is a promising new potential treatment for inflammatory bowel disease and colon cancer [20].

*Bilophila wadsworthia* is one of the well-known bacterial species that produces hydrogen sulfide from sulfite [21]. This opportunistic pathogen quickly digests bile and was isolated from fecal and appendices specimens [5]. It may be possible to treat these medical conditions by focusing on the gut bacteria *B. wadsworthia* and sulfate-reducing bacteria (SRB) that produce hydrogen sulfide [20]. Isethionate (2-hydroxyethanesulfonate) is an essential source of sulfur produced mainly from the deamination of taurine. Taurine is the second most abundant free amino acid in the human ileum and proximal colon [22]. A glycal radical enzyme called isethionate sulfite-lyase is responsible for breaking the C–S bond in isethionate to produce acetaldehyde and sulfite, mainly present in *B. wadsworthia* [21,23]. *B. wadsworthia* is a Gram-negative, colonial bacteria ranging from 0.7 to 0.9 mm in width and from 1.0–10.0 mm long [24]. Isethionate sulfite-lyase (Isla) is a member of the glycyl radical enzyme (GRE) superfamily, which carries out a variety of chemical processes in anaerobic environment [25]. The first crystal structure of the Isla of *Desulfovibrio vulgaris Hildenborough* was reported (PDB ID: 5YMR) having substrate attached in the active site, which investigates the mechanism of bacterial degradation of organosulfonates and toxic H_2_S production [26]. The active site of this enzyme is constituted in a ten-stranded α/β barrel design that is shared by all reported GREs. This architecture includes the Cys loop and the Gly loop; those two loops are named for catalytically essential residues.

Each GRE has a specific [4Fe-4S] activase (IslB for the Isla GRE) that connects a radical on a glycine residue on the Gly loop to activate it. This glycyl radical is believed to take a hydrogen atom from a conserved cysteine residue on the Cys loop during the chemical cycle, creating a catalytic thiyl radical. To create a substrate radical that can then rearrange into a product radical, this thiyl radical removes a hydrogen atom from the substrate [27]. Recently, a full structure/function analysis of Isla, in which the native Isla structure, the Ise-bound structure of Isla from *B. wadsworthia* ia (Isla) at 2.70 Å resolution, was reported. Isla is dimeric, as is the case for the fundamental structure of all characterized GRE eliminates, and each monomer has a hidden active site that is centrally located within a barrel made up of two five-stranded half-b barrels that are antiparallel to one another (b1–10) and are encircled by an α-helix [28]. It is reported that this hidden active site protects radical species from solvent quenching. The Cys loop and the Gly loop are two adjacent and catalytically significant loops in the active site. A C-terminal glycyl radical domain with the Gly loop and its conserved Gly residue is present in all GREs (Gly805 in Isla). The conserved catalytic Cys residue is part of the Cys loop (Cys468 in Isla) and shows the location of Gly805 and Cys468, which are 5.2 Å apart and capable of transferring radicals to one another, as well as producing a brief thiyl radical that kickstarts catalysis on the substrate [20]. 

Recently, the structure of Isla enzyme was reported in apo form and in complex with its substrate which describe the active site of Isla and substrate-binding mechanism. Interestingly, no drug molecule is reported for this potential drug target, which diverts our attention towards exploring the inhibitory potential of diverse chemical entities for Isla. With our interest in designing druglike molecules using in silico methods [29,30,31,32], we carried out this current project focusing the identification of small inhibitor candidates by applying structure-based computational drug design protocol.

## 2. Materials and Methods

### 2.1. Protein Crystal Coordinates Selection

The reported X-ray diffraction structures of *Bilophila wadsworthia*, isethionate sulfite-lyase enzyme were retrieved from RCSB protein Data Bank (https://www.rcsb.org/: accessed on 29 August 2022), having a substrate (2-hydroxyethylsulfonic acid) attached in the active pocket (PDB-ID: 7KQ3) and in free state (PDB ID: 7KQ4) [20]. The missing residues in the 3D structure of the isethionate sulfite-lyase were modeled by Molecular Operating Environment version 2022.02 (MOE) [33] embedded loop modeler algorithm using Amber14:EHT (Amber ff14SB combined with EHT) forcefield [34,35]. The start and the end (C–N terminals) residues of isethionate sulfite-lyase were charged, and the missing hydrogens were added to the protein residues by MOE Quickprep tool. Any deficiencies such as missing parameters in the forcefield, such as missing atom types, van der Waals, angle parameters, bonds information, and residues chirality, were added.

### 2.2. Molecular Docking and ADME Prediction

The in-house database having ~800 compounds (isolated from natural products and their synthetic derivatives) was selected to screen against the isethionate sulfite-lyase protein. The putative interactions of those compounds with the active site residues of sulfite-lyase were investigated by molecular docking method using MOE. The refined structure of the isethionate sulfite-lyase protein (PDB ID: 7KQ3) with attached substrate (2-hydroxyethylsulfonic acid) in the active pocket was selected for docking. The docking protocol was validated by redocking the substrate in the protein’s active site. A short minimization of enzyme was performed with Amber14:EHT forcefield until RMS gradient of 0.1 kcal/mol/Å was obtained before proceeding with the docking. The “Triangle Matcher” algorithm was used to place the substrate at 100 possess based on the bond rotation and scored by the “London dG” scoring function. The final 30 refined possess from the substrate was selected based on the “GBVI/WSA dG” algorithm. The RMSD was calculated between the cocrystallized and docked substrate. Before docking, the geometry of small ligands in the database was optimized by calculating the partial charges and adding hydrogens using MMFF94x forcefield in MOE. Thirty docked poses of each compound were preserved, and the optimal docked orientation of each compound was chosen based on the docking score and maximal binding interactions. The pharmacokinetics (absorption, distribution, metabolism, and excretion, ADME) of hits obtained after docking were estimated by SwissADME service (www.swissadme.ch: accessed on 9 November 2022). To calculate the pharmacokinetics, the SMILE format of each compound was submitted to SwissADME, and their ADME, drug similarity, and medicinal chemistry characteristics were analyzed [36].

### 2.3. Molecular Dynamic Simulations

Molecular dynamics (MD) simulations were conducted in the substrate-bound activated state (7KQ3), ligands-bound inhibited states (CP1–CP5) and in free state (7KQ4) to investigate the structural dynamics of sulfite-lyase. Then, 7KQ3 was selected as the reference activated state of sulfite-lyase, while 7KQ4 was used as an apo or free state, and the most potent ligands obtained in docking with the sulfite-lyase (CP1, CP2, CP4, CP5, CP8, and CP9) were simulated using AMBER22 engine [37]. The forcefield ff19SB [38] (amino-acid-specific forcefield) was used for protein, and the protein coordinates and topology were created using AMBER22’s LEaP module. The geometry of small compounds was optimized by General Amber Force Field-2 (GAFF2) [39] to calculate the AM1-BCC [40] charges. The missing forcefield parameters for small molecules were generated with the AMBER22 embedded tool Parmchk2 [41]. A 1 Å grid Na^+^ and Cl^−^ (~0.1 M) monovalent OPC ions were added to each system for neutralization. The AMBER22 LEaP module was used to add the missing hydrogen to the protein. In MD simulation, the systems were solvated with an OPC (optimal point charge) water model with a 10 Å buffer distance in an octahedral box. To boost parallel scalability, the long-range electrostatics were calculated using particle mesh Ewald molecular dynamics (PMEMD) [42] integrated with AMBER22. The protein minimization preceding MD was achieved in two steps. In the first phase, the steepest descent algorithm minimization was performed in 10,000 steps with restraints on the protein residues, followed by the conjugate gradients minimization in the second phase with 5000 steps minimization, respectively [43]. To heat up the systems, the temperature was increased steadily from 0.1 to 300 K at 400 ps with weak restraints applied to the protein utilizing the Langevin thermostat [44] of temperature. The Langevin thermostat and the collision frequency of 2.0 ps-1 were used to adjust the protein kinetic energy of harmonic oscillations for dynamic propagation. In 400 ps, the protein density was adjusted with the heating protocols. Before the MD production run, equilibration was carried out utilizing the 300 K temperature using an NPT ensemble for 2000 ps. The hydrogen atoms were treated with the shake algorithm with a 2 fs time step [45]. The systems were subjected to a cutoff of 8 Å in order to compute the long-range electrostatics. The equilibration step protocols were further used for a 100 ns production run which included the reference activated system (7KQ3), free state (7KQ4), and the top hits (CP1, CP2, CP4, CP5, CP8, and CP9). Each system’s output trajectory was generated at a simulation time of 10 ps.

### 2.4. Post Dynamic Evaluation

In order to examine the stability of each system’s 100 ns long output trajectory, AMBER22’s CPPTRAJ module [46] was employed. Each system’s 11,000 frames trajectory was used to construct the Cα-atoms based root mean square deviation (RMSD). The initial frame of the trajectory was used as a reference coordinate to compute the RMSD. The average of the RMSD over time was computed to comprehend the fluctuation in the RMSD. The stability of the ligand in the protein active pocket during simulation was validated by using all atoms of the small compounds to compute the RMSD. The small molecules’ minimization coordinates were considered the root dimensions for ligand RMSD.

### 2.5. Assessment of Structural Dynamic Changes 

Using the RMS-histogram analysis incorporated in CPPTRAJ, it was helpful to quantify the total RMSD changes in the isethionate sulfite-lyase structure as converged. The root mean square fluctuation (RMSF) analysis, which denotes the flexibility of every residue in the system, was used to evaluate the residual fluctuations in isethionate sulfite-lyase during simulation. The radius of gyration (Rg) analysis was carried out for each system to investigate the compactness of protein in the activated state (7KQ3), free state (7KQ4), and ligands attached states (CP1, CP2, CP4, CP5, CP8, and CP9). The shared center of gravity was evaluated to obtain Rg values during simulation.

### 2.6. Principal Component Analysis

The principal component analysis (PCA) implemented in the CPPTRAJ was used to analyze the dominating movements in the isethionate sulfite-lyase structure during the simulation timeframe in the free state (7KQ4), reference activated state (7KQ3), and inhibited states (CP1, CP2, CP4, CP5, CP8, and CP9) [46,47]. The ten motion modes were used to generate the coordinate covariance matrix, and each system’s 3D positional coordinates were obtained. From each system’s 11,000-frame trajectory, the eigenvalue and eigenvector of the Cα-atoms were determined, and the main components were constructed by diagonalizing the results (PCs). The eigenvalues display the motion amplitude, while the eigenvectors depict the direction of movement in the PCs. Fractions were used to represent the eigenvectors that were produced for each system. Highest degree of variance in PCA analysis is seen in PC1 and PC2; therefore, PC1 and PC2 were plotted against each other to study their movements.

### 2.7. Hydrogen Bonds Analysis

Using the hbond feature of CPPTRAJ, the interactions between isethionate sulfite-lyase with the substrate (hydroxyethylsulfonic acid) and with the top selected inhibitor candidates (CP1, CP2, CP4, CP5, CP8, and CP9) were examined throughout the simulation [48]. The number of hydrogen bonds between the substrate/ligand and the target that were created throughout the simulation, as well as their average life span, distance, and angle, were calculated using the 100 ns long trajectory of each system. The protein-substrate/ligand electron acceptor and donor atoms were set to a 3.5 Å cutoff distance and bond angle of 120°. 

### 2.8. Binding Free Energy Calculation

The molecular mechanics/generalized born surface area (MM/GBSA) approach was used to determine the binding free energy between the isethionate sulfite-lyase and substrate/ligands bound to the active site [49,50,51]. The binding free energy of the ligands was calculated using the final 2200 frames (20 ns) of every output trajectory. Using the 2 Å solvent probe and the mbondi3 radii, the topology of each system was refined. Following Equation (1), the binding free energy (ΔG_bind_) was computed. The protein–ligand complex energy (ΔG _R + L_) was deducted from the receptor (ΔG_R_) and ligand (ΔG_L_) individual energies to calculate the binding free energy.
(1)$$\begin{array}{c} {\Delta G}_{\mathrm{bind}}={\Delta G}_{R+L}\;–\;\left({\Delta G}_{R}+{\Delta G}_{L}\right) \end{array}$$

Following Equation (2), the individual energies that made up each system energy in Equation (1) were calculated.
(2)$$\begin{array}{c} G=E_{\mathrm{BOND}}+E_{\mathrm{VDW}}+E_{\mathrm{EEL}}+G_{\mathrm{PB}}+G_{\mathrm{SA}}–\;T_{\mathrm{SS}} \end{array}$$

Dihedrals and angles (G_BOND_), van der Waals energies (G_VDW_), electrostatic energies (G_EEL_), polar and nonpolar solvation energies (G_PB_ and G_SA_), and temperature (T) with the solute entropy (SS) are the contribution energies of each system in Equation (2). With the exception of the polar solvation energy, which was determined using the LCPO technique, all energies were calculated in kcal/mol, and the protein’s surface area was determined in Å^2^. The direct sum of the stated electrostatic energies was used to compute the total energy of each system.

### 2.9. Data Analysis 

MOE was used to analyze the protein–ligand interactions, and visualization was carried out by VMD [52], Pymol [53], and Blender [54]. Origin was used to create the plots [46]. CPPTRAJ of AMBER22 was used to build the structural ensembles with the lowest energy conformation.

## 3. Results and Discussion

### 3.1. Structure of Isethionate Sulfite-Lyase and Molecular Docking

The X-ray diffraction structure of the isethionate sulfite-lyase from *Bilophila wadsworthia* (PDB ID: 7KQ3) with the substrate (2-hydroxyethylsulfonic acid) bound in the active site is depicted in Figure 1. The newly identified mechanism of substrate binding reveals that substrate molecule (2-hydroxyethyl sulfonic acid) mediates strong hydrogen bonding (H-bonds) with Glu470, Arg189, Gln193, Arg678, and Cys468. In addition, Arg189 and Arg678 also provide ionic interaction to the substrate. This newly elucidated binding mechanism of substrate helped us to design drug molecules which target these residues or mimic binding mechanism of substrate. The binding mode analysis of compounds reflect that compounds **1**, **2**, **4**, **5**, **8**, and **9** establish good interactions with few of those residues. Compound **1** formed multiple H-bonds with Arg678, Arg189, and Gln193, while Thr312 also provides a H-bond to CP1. Among the selected hits, this compound exhibits the highest docking score, i.e., −6.62 kcal/mol. Compound **2** formed H-bonding with Glu470, Gln544, and Ala484, however, Arg678 provided π–cation interaction to this compound. The docked view of compound **4** shows H-bonding between CP4 and Tyr587, Arg189, and Tyr485. Among all the selected hits, compound **5** formed the highest number of H-bonds with Thr185, Tyr587, Arg189, Gly483, Arg678, and Cys468. These molecular interactions between CP5 and Isla show greater affinity of this compounds towards Isla. The compounds **8** and **9** also formed significant interactions with Cys468, Tyr587 and Tyr587, Gly586. Additionally, CP8 also formed π–cation with the side chain of Arg189. These molecular interactions are responsible for the good docking scores of these compounds, which are in the range of −6.62 to −5.35 kcal/mol. The detailed docking interactions of the selected compounds with the isethionate sulfite-lyase active site residues are tabulated in Table 1.

### 3.2. ADMET Analysis 

Developing an ordinary chemical entity into a suitable medicine depends on several characteristics, including efficacy, safety, and its selectivity for the biomolecular target. It is possible to evaluate a drug’s pharmacodynamic (PD) and pharmacokinetic (PK) qualities by looking at a number of factors, including high potency, affinity, absorption, distribution, metabolism, excretion, and toxicity (ADMET). These characteristics determine solubility (LogS) and lipophilicity (LogP). Several parameters were predicted and evaluated, which are summarized in Appendix A. The key physiological and molecular properties revealed that the molecular weight (MW) of selected anti-Isla molecules (CP1, CP2, CP4, CP5, CP8, and CP9) ranges from 157–255 g/mol and these molecules have no blood–brain barrier (BBB) penetration. The number of rotatable bonds (NRB) and predicted polarity (TPSA) of the selected compounds ranges from 1–5 and 89.1 to 128.32 Å^2^, respectively. TPSA value ≤ 140 Å^2^ signifies high likelihood of good oral bioavailability according to Veber’s rules. The number of hydrogen bond acceptors (HBAs) and donors (HBDs) in the chosen potential inhibitors are 2–5 and 1–2, respectively. In addition, those selected compounds obey the Lipinski rule of drug-likeness with hydrogen bond acceptors (HBA) < 10. As described by the Ghose rule, oral bioavailability is determined as the key factor and measured in the molar refractivity (MR) values with range of 40–130. The selected compounds MR values are within the range of the Ghose filter rule. For the chosen chemical compounds, Lipinski’s rule, Ghose filter, Vebers’s, Egan’s, and Muegge’s rules were calculated to predict their drug-likeness. All the selected compounds show no violation in Lipinski’s rule, Ghose filter, Vebers’s, and Egan’s rules, except a single violation of (WLOGP <−0.4) Muegge’s rules was shown by CP1, CP2 and CP4. The selected compounds (CP1, CP2, CP4, CP5, CP8, and CP9) show no violation of three drug-likeness rules, suggesting that they can be subjected to further investigations as potential inhibitors of Isla.

The selected compounds show high gastrointestinal (GI) absorption and do not show the possibility of blood–brain barrier (BBB) penetration. In addition to that, the selected compounds are described as nonsubstrate for P-glycoprotein (P-gpS). For the cytochrome P450 (CYP) monooxygenase enzymes, the selected compounds are noninhibitors, demonstrating the safe clearance prospect after the drug’s metabolism is ensured within the human body. Moreover, the skin permeability (Log Kp) is within the range of −5.96 to −7.61 cm/s (more negative values show less permeability). The bioavailability score of the selected compounds is reported to be 0.55, which signifies good pharmacokinetic properties. The selected compounds have no PAINS alert calculated by the SwissADME. The synthetic accessibility score calculated (2.03–3.09) signifies that these selected compounds can easily be synthesized.

### 3.3. Molecular Dynamic Simulations

#### 3.3.1. Evaluation of Structural Stability Via Root Mean Square Deviation

The C-α-atoms-based RMSD of Isla in free state (7KQ4), isethionate-bound Isla (7KQ3), and in complex with six selected putative inhibitors (CP1, CP2, CP4, CP5, CP8, and CP9), were calculated and are depicted in graphs (Figure 2). To understand the stability of the system overtime, the first frame of the trajectory was chosen as reference coordinates in RMSD calculation. The average RMSD of 7KQ4 is 1.53 ± 0.21 Å. The RMSD of 7KQ4 increased gradually from the start of the simulation till 100 ns (1.9 Å average) and remains constant afterwards (till 200 ns). The global trend of ligand-bound protein complexes is substantial variations in the backbone RMSD, with most systems representing higher average value than the 7KQ4 (except CP8, which showed mean RMSD 1.33 ± 0.18 Å), indicating conformational shift induced by the ligand attachment. The reference complex 7KQ3 resulted in high-magnitude oscillations around the RMSD (1.95 ± 0.38 Å) throughout the simulation, as compared to 7KQ4, and stayed stable after 120 ns till the end. The CP1–Isla complex depicted a similar pattern with the 7KQ3 in the RMSD profile, reaching a peak of 1.60 Å average during first 30 ns simulation time. A sudden rise from the start of the simulation was observed in the system’s backbone RMSD, and then the system’s continual RMSD was maintained for most of the simulation time, involving a fluctuation period of 50 ns, 70 ns, 120 ns, and 160–180 ns. For the CP2-bound Isla, an equilibrium state was accomplished after 20 ns. Before this, a sudden rise in the RMSD was observed in the complex (1.01 Å to 1.75 Å), and then the system underwent oscillations around the mean value of 1.7 Å till 90 ns. From 90–100 ns simulation time, the conformation adjustments were observed to stabilize the ligand in the active site of Isla (mean RMSD 1.55). Following that, a stabilized protein–ligand complex formation was noticed with a constant RMSD of ~1.55 Å for most of the remaining simulation time for the CP2–Isla complex. In the CP4–Isla complex, the structural changes were observed in the protein upon ligand bindings during the first 100 ns, with RMSD ranging from 0.98 Å to 1.98 Å. The system RMSD was then stabilized with average 1.75 Å till the end of simulation. In case of CP5–protein complex, the RMSD gained equilibrium during first 70 ns time with RMSD ranges from 0.95 Å to 2 Å, facing persistent mobility. The CP5 complex maintains the RMSD with an average of 2.09 Å till the end of the simulation, although the former system exhibited a slightly higher average RMSD (1.87 ± 0.33 Å) than the other selected compounds complexes. The average RMSD of CP8–protein complex is 1.43 ± 0.18 Å, which faced significant mobility in the initial 20 ns (RMSD: 0.88 Å to 1.28 Å); however, from 80 ns to 160 ns, the system increases the RMSD~1.72 Å, and it indicates that the ligand binding caused conformational perturbations in the protein’s structure. The system retains the RMSD with an average of 1.47 Å till the 200 ns. The CP9-bound Isla underwent significant flexibility during the initial 45 ns (gain in RMSD of 2.07 Å) and maintained the average RMSD of 1.70 Å till the completion of simulation. These observations showed that most of the inhibited complexes attained a stable configuration throughout the simulation with minor conformational transitions. In addition, the significance of our simulation run is indicated by minimal RMSD variations during the simulation.

#### 3.3.2. Evaluation of Ligand Stability in the Active Site of Isla

In order to ascertain the stability of ligands, the RMSD acquired from protein fitting its ligand was studied across simulation of 100 ns. A high RMSD value indicates that the ligands underwent significant fluctuation during the simulation (Figure 3). The estimated mean RMSD of our selected putative inhibitors (CP1, CP2, CP4, CP5, CP8, and CP9) (>3.5 Å) is less than the reference system 7KQ3 (i.e., 3.75 ± 0.65 Å). The substrate in 7KQ3 remains less stable within the binding pocket for most of the simulation time. However, a significant gain of RMSD was noticed in the period of 200 ns simulation time (RMSD: 1.62 Å to 5.03 Å). The CP1 maintains stable interactions with the active site residues of Isla, indicated by a stable RMSD profile with only slight conformational transitions. The mean RMSD for the CP2 is 1.91 ± 0.004 Å. The RMSD of CP2 was sustained around 2.75 Å during first 60 ns. This ligand reached a peak RMSD value of 2.6 Å till the end of the simulation. In case of CP2, a continuous decrease in RMSD was observed in 200 ns simulation time (2.87–1.59 Å). The mean RMSD computed for the CP2 is 2.27 ± 0.003 Å, where the ligand underwent significant flexibility in the active site. The average RMSD reported for CP4 is 1.95 ± 0.002 Å, where the ligand shows a stable behavior (minor fluctuations in the RMSD) and retains a 1.75 Å RMSD in most of the simulation time. The protein active pocket experienced significant conformational changes with CP5 during first 80 ns (0.6–4.1 Å). The CP5 stabilizes the interactions in the active pocket and retains the RMSD at a 3.8 Å average till the rest of the simulation time. An increase was observed in the average RMSD of CP5 (3.45 ± 0.006 Å). CP8 shows stable behavior retaining the RMSD at 3.45 Å in most of the simulation time. The average RMSD reported for CP8 is 3.35 ± 0.001 Å. Similarly, the CP9 RMSD pattern is similar to the CP5, where the first 80 ns simulation time shows a steep increase in the RMSD (2.14–3.93 Å), and the ligand obtains stability after 80 ns in the protein active pocket till the end of the simulation. The mean RMSD of the CP9 system observed is 3.35 ± 0.006 Å. These findings suggested that the chosen ligands showed good interactions during the simulation and formed a stable complex with the Isla protein.

#### 3.3.3. Thermal Stability Analysis of the Protein

The root mean square histogram (RMS-hist) analysis during simulation was used to examine the structural conformation of Isla (in its free and inhibited states) (Appendix A). The thermally stable conformation of apo–Isla (7KQ4) was observed in 1.4 Å, where the RMSD of the overall system was increased by ~2.1 Å. The stable conformation of reference complex (7KQ3) was calculated in the 2.4 Å RMSD, and with the overall RMSD of 2.7 Å. The inhibited complexes showed different patterns of protein conformation during the simulation time, where the ligand attachment causes structural deformation in the protein from its active state. The equilibrated conformations of CP1 and CP2 bound complexes were observed in 1.6 Å RMSD, with overall RMSD of 2.4 Å and 2.0 Å, respectively. The thermal stable conformation of CP4 and CP9 bound complexes are at 1.3 Å. In addition, the overall RMSD of those complexes fluctuated to 2.4 Å during the simulation. The stable conformation of protein in CP5-bound complex was observed in 2.1 Å, with minor overall RMSD fluctuation (2.4 Å). In the CP8 complex, the protein equilibrium is observed at 1.5 Å and the overall RMSDF of the protein did not shift much (1.8 Å). The RMS-hist analysis revealed that the ligand-inhibited complexes experience various structural variations due to the ligand attachment, thus changing the protein RMSD, which concludes that the selected inhibitor candidates have steady attachment and influence the dynamics of the protein structure.

#### 3.3.4. Residual Flexibility of Isla

We computed the root mean square fluctuation (RMSF) analysis for 7KQ4, 7KQ3, and six putative Isla-inhibitor complexes to examine the divergence of a specific region of protein from the mean location (Figure 4). Each residue’s RMSF variations show the degrees of mobility that each residue has acquired. Because of this, a residue or group of residues that experience higher levels of RMSF upon ligand attachment likely have greater flexibility, which increases the probability that they may interact with the ligand molecule. Smaller RMSF deviations consistently indicate lower mobility and, hence, less opportunity for interaction with the ligand molecule. The mean RMSF values of the ligand’s attached systems were reported to be high as compared to 7KQ4 (apo–Isla) (0.74 ± 0.020 Å). The protein residues region from 202 to 234 in the 7KQ4 showed highest fluctuation (2.50 Å) due to a loop conformation. Residues towards the N-terminal end showed higher fluctuations in the apo–Isla as well as all ligand-bound Isla complexes. The remaining protein shows a stable conformation with RMSF <1 Å. The reference system (7KQ3) shows high fluctuation in the 202–234 residues position (3.93 Å) with average RMSF of 0.85 ± 0.021 Å. The interactive residues of the active site including the 100–150, 320–340, 489–532, and 600–649 regions show fluctuation of ~1.61 Å due to the interaction of the substrate. In the CP1 complex, the region of 108–243 residues has the most deviation where the interactive residues of the protein fluctuate greater than 3 Å, suggesting good interaction of CP1 with the residues of active pocket, while residues 661–678 fluctuated by ~2 Å due to the involvement in the action with CP1. The rest of the protein remains stable during the RMSF analysis with a mean RMSF of 0.81 ± 0.022 Å. In the case of CP2, high fluctuations were observed (2 Å) in the region of 214–249, 324–345, 494–533, and 662–674 residues, including the protein–ligand interactive residues. The average RMSF of the CP2 complex shows a decrease in the RMSF value of 0.67 ± 0.018 Å. The fluctuation above 2 Å was observed in the CP4 complex in regions of 202–234, 324–382, and 649–678. These residues include the active site residues, which interact with CP4. The average RMSF reported for the CP4 complex is 0.80 ± 0.020 Å. Similarly, the average RMSF calculated for the CP5 system is 0.79 ± 0.023 Å, where the interactive residues regions (108–166, 202–234, 327–376, and 598–662) of the protein active pocket fluctuate up to ~2 Å. The regions of 108–166 and 202–234 residues show a fluctuation of 2 Å, also including the N-terminal region of the CP8 system, where the average RMSF of the system remains 0.74 ± 0.017 Å. The regions involved in interactions with the CP8 also include 323–344 and 659–680 residues with a fluctuation of 1.6 Å during the simulation. Finally, the CP9 complex has high mobility in the 109–142 and 211–240 residues, fluctuating by 3.15 Å and 3.55 Å, respectively. The 327–342 residues region in the CP9 complex shows a fluctuation of 1.57 Å with the overall RMSF of 0.72 ± 0.023 Å. These findings demonstrate that a number of the protein’s nonbonded residues showed higher mobility, indicating their potential of interaction with ligands; it also demonstrates that the ligands are well-fitted in the binding pocket of Isla.

#### 3.3.5. Analysis of Protein Compactness 

The structural compactness of Isla before and after the ligand binding was examined through radius of gyration (Rg) analysis, and the resulting trajectory was graphed (Figure 5). The apo–Isla (7KQ4) and isethionate-bound Isla (7KQ3, control) had average Rg values of 26.640 ± 0.002 Å and 26.86 ± 0.001 Å, respectively. The computed average Rg values for the ligands–Isla complexes are high, except the CP2 and CP9, which showed somewhat lower mean Rg values (26.64 ± 0.001 Å and 26.63 ± 0.001 Å, respectively) than the control. The Rg value estimated for apo form increased slightly (0.2) until 70 ns; however, after that, the value remained relatively steady, in the range of 26.66 Å until the simulation ended. On the other hand, ligand binding resulted in a slight rise in Rg value due to the occupied intramolecular space of Isla by the bound ligand. For instance, the binding of an isethionate disrupted the tight packing of protein secondary structure components, as shown by a slight uptick in Rg value (particularly between 0 ns and 110 ns). A steady Rg was then measured, with a small variation around the mean value, indicating the presence of a stable protein–ligand complex. The variation in Rg of CP1 complex during the simulation can be divided into two sections. In the first 100 ns, the Rg was increased slightly (26.56–26.82 Å), while in the second section, the system retains a steady conformation with an Rg value of 26.82 Å till the termination of the simulation. In the case of the CP2 complex, the Rg fluctuates (26.49–26.64 Å) at the first 20 ns. Afterwards, the Rg of this complex was enhanced to 26.63 Å average across the simulation time, suggesting a stable complex. A sudden jump in the Rg values was observed in the CP4 complex (26.50–26.90 Å) during the first 40 ns simulation period. A compact nature of CP4–Isla complex was observed after 40 ns, evident by the stable Rg value around 26.96 Å. The CP5 complex showed a similar pattern of Rg with high jumps during first 40 ns (16.47–26.78 Å), resulting in an unstable conformation. Later, the complex obtains a tight packing confirmed by the stable Rg values (26.84 Å mean) for the remaining simulation time. The CP8–Isla complex reflects a compact nature during the 200 ns simulation time, with average 26.73 Å Rg value in most of the simulation time, having a slight increase of 0.25 Å at the start of the simulation. The CP9 complex has an instability period in the first 20 ns simulation. In contrast, the ligand attachment led to a compact protein complex with Rg value around 26.63 Å till the 200 ns simulation time. Overall, these findings indicate that the protein is tightly packed after ligand binding, supporting the formation of a stable protein–ligand complex. 

#### 3.3.6. Protein Principal Movements

The Cα-based principal component analysis (PCA) was used to calculate the total Isla domain mobility during the simulation. The eigenvectors of the covariance matrix, indicated by the coinciding eigenvalues, reflect the entire combined motion of the Cα atoms in the Isla, and the PCA helped in their finding. The number of eigenvectors with high eigenvalues often represents the overall coordinated motion of the Isla protein in relation to its function. The significant movements that the protein structure endured over the simulation time were computed in fractions using the ten eigenvectors, depicted in Appendix A. The first three eigenvectors of each system exhibited the intense movements caused by the residues, whereas the remaining eigenvectors simply depicted localized motion. The highest motions observed in the Isla during the simulation time were in the substrate-bound Isla (7KQ3), i.e., 42%. The apo–Isla (7KQ4) shows 29% of the dominant motion during the 200 ns trajectory analysis. The compounds CP1, CP5, CP8, and CP9 showed higher motions (35%, 36%, 36%, and 38%, respectively) in the protein structure as compared to 7KQ4. In addition, the CP2 (25%) and CP4 (27%) reflect restricted motions in the protein due to the ligand binding. These movements observed in the selected ligands confirmed the ligand stability in the protein active pocket and protein structural conformation caused by the ligands. 

The flipping-over conformations of the protein with attached inhibitors and free state were analyzed by plotting the PC1 over PC2. The dots indicated the conformational switching over for each frame during the simulation timeframe. The simulation frames’ color density varied from blue to red, with red signifying the protein’s persistence at that position (Figure 6). A high divergence was observed in the flipping over conformations of the apo–Isla (7KQ4), 7KQ3 and selected inhibitors systems. The 7KQ4 starts from the 70 x-axis and ends at the value of −60 y-axis, having three density-stabilized regions. Further, the 7KQ3 has two conformational divergences in the simulation pattern, which stay both at the positive sides of the x- and y-axes. The selected inhibitors show a diverged pattern of motion over the simulation period as compared to 7KQ4 and 7KQ3. These results indicate that the behavior of Isla changes when the ligand is attached with the active pocket, and these inhibitors bring conformational changes in the protein dynamics, which may alter the normal function of Isla. 

The direction and intensity of the motions observed during simulation in the Isla are presented in the porcupine plots in Figure 7. In the figure, the length of the arrows reflects the amount of motion created in the residue’s region, while the direction of the arrow denotes the direction of motion. The apo–Isla (7KQ4) shows outside motions from the active site position; thus, protein acquires an open conformation. Parallel to that, the 7KQ3 having a substrate attached exhibits movements towards the active site, as a resulting protein adapts a closed conformation. The selected ligands complexes revealed diverse motions from the active site region of the protein, which reveals structural changes in the protein due to the inhibitor attachment. Overall, the inhibitors-bound Isla adjusted in a closed conformation during the 200 ns simulation time, which confirms the ligand’s stability in the protein’s active pocket. 

#### 3.3.7. Hydrogen Bond Analysis 

Intermolecular hydrogen bond formation is critical for the persistence and directionality of a protein–ligand complex. The affinity of a ligand towards the binding site of the protein may be measured using intermolecular hydrogen bonds. The hydrogen bonds established between Isla and chosen ligands were recorded and plotted during the simulation to investigate their formation and breakdown (Figure 8). The substrate in Isla (7KQ4) strongly interacts with GLU470, ARG678 (3x), ARG189 (5x), TYR485 (3x), ARG678, TYR587(3x), GLN193, and CYS468, with life span of 52.43%, 42.34%, 40.67%, 36.45%, 24.50%, 22.26%, 16.70%, and 15.89%, respectively. The residues which help in the stabilization of substrate (>10% occupancy) in the active pocket include ARG678 (3x), THR312 (3x), GLU470, GLN193 (2x), VAL680, THR312, TRP374 (3x), TYR587, TRP374, and GLY483. In the case of CP1, strong hydrogen bonds were observed with ARG189 (3x), THR312, TYR587, GLN193, and GLU589 (2x) with calculated bond life of 60.24%, 52.20%, 44.53%, 34.99%, and 22.41%, respectively. In addition, several residues (GLY311 TYR485 (3x), GLN193 (2x), PHE682, VAL680, ARG189, ARG678 (5x), and GLU589 (2x)) in the active site support the fitting of CP1 (>10% occupancy). The CP2 forms very strong interaction with TYR479 (78.58% occupancy), ASN675 (75.73% occupancy), and ALA484 (21.40% occupancy) residues in active pocket. Besides these residues, the helping residues which firm the CP2 stability in the Isla pocket comprise TYR587, GLN544, TYR479, and GLY482 (>10% occupancy). The ligand CP4 form strong hydrogen bonds with TYR587 and SER481, with the bond occupancy rates of 61.47% and 15.32%, respectively. The TYR479, GLN544, GLY483 (2x), TYR587, and THR185 (>10% occupancy) contribute to the attachment of the inhibitor–protein complex. The Isla residues which form robust interactions with the CP5 involve GLU589 (2x), ARG189 (2x), and TYR485 with high bond occupancy of 96.38%, 16.20%, and 14.11%, respectively. Furthermore, the supporting residues of the protein involved with the ligand are TYR587 (2x), ALA484 (2x), TYR485 (3x), ARG678 (4x), GLY483, ARG189 (6x), CYS468, TRP374 (2x), and THR312 (2x). The strongest interactive residues with the CP8 comprise GLU589 (2x), GLY483, GLN193, and TYR587. The bond life of these residues is 63.64%, 39.57%, 24.07%, and 10.00%, correspondingly. The Isla active site residues which play a role in the equilibrium of the ligand are ARG189 (2x), THR185 (2x), TYR587, GLN193, TYR479 (2x), TYR485 (3x), and GLU470 (2x) (>10% occupancy). Finally, the CP9 interacts with the THR185, GLY483, GLN193 (2x), SER481, GLU470, and ARG189, forming a stable hydrogen bond having lifeline of 60.24%, 59.63%, 18.96%, 17.22%, 12.60%, and 12.57%, respectively. The residues which impact the strengthening of CP9 (>10% occupancy) in the active site include GLU470, ALA484, TYR587 (2x), SER481, GLN193 (3x), THR185, ARG189 (2x), TYR485 (4x), GLY483 (2x), ARG678 (4x), GLN544, and TYR479. All the chosen inhibitors displayed substantial H-bond interactions with the Isla target site, indicating their potential as effective disease therapy alternatives. Figure 8 and Appendix A show a comprehensive hydrogen bond analysis.

#### 3.3.8. Free Energy Calculations 

To compute the binding affinity between selected hits and Isla, MM-GBSA calculations were performed and are provided in Table 2. Moreover, various free energy constituents (van der Waals interactions (∆E_VDW_), electrostatic interactions (∆E_EEL_), polar solvation energies (∆E_GB_), and nonpolar solvation energies (∆E_SURF_)), which contribute to the total energy (∆G_TOTAL_) were also computed. MM-GBSA simulations found binding energy discrepancies between the substrate and the selected inhibitors. The substrate attached in the 7KQ3 shows a binding free energy (∆G_TOTAL_) of −15.53 kcal/mol. As compared to substrate energy, all of the chosen inhibitors exhibited high binding free energies. The highest binding free energy is exhibited by CP4 (−34.27 kcal/mol), followed by CP5 and CP2 (−30.83 and −30.17, respectively). The ∆E_VDW_ of CP4, CP5, and CP2 (−45.56, −36.43, and −30.88 kcal/mol) is high compared to the substrate molecule (−10.52 kcal/mol). The surface area of those complexes also decreased (−5.50, −4.60, and −4.11 Å^2^) compared to the reference substrate (−3.23 Å^2^), while ∆E_GB_ of these compounds (31.14, 67.78, and 39.83 kcal/mol) was increased significantly compared to the reference (−67.00 kcal/mol). The CP1 depicts binding free energy of −26.85 kcal/mol where the protein surface area did not change (−3.23 Å^2^) compared to the reference. The ∆EVDW (−23.91 kcal/mol), ∆EEEL (−48.61 kcal/mol), and ∆EGB (48.90 kcal/mol) also increased in the CP1 system. The CP8 and CP9 systems exhibit binding energy of −21.54 kcal/mol and −23.22 kcal/mol, respectively, with the Isla. These two systems dominate in the ΔE_vdW_, ΔE_elec_, and ΔE_SURF_ free energy components compared to 7KQ3 (Table 2). Overall, the results demonstrated that the chosen inhibitors interact with increased affinity and confer energetically favorable binding as compared to the reference (substrate), and therefore may serve as potential inhibitors of Isla.

## 4. Conclusions

Isethionate sulfite-lyase (Isla) produced by Gram-negative bacteria, *Bilophila wadsworthia,* can break the mucous lining of the colon. As a result, several pathological conditions may occur, including colorectal cancer, inflammatory bowel diseases, and colitis. Recently, a three-dimensional structure of this interesting target was determined in complex with its substrate, and the binding mechanism of this enzyme was explored [20]. However, until now, none of the drug molecules that inhibit its function have been reported. This opens a door to explore the potential molecules that can inhibit the function of this enzyme. With our interest in designing novel and potential drug-like molecules, we tested the efficacy of our in-house database of compounds against the Isla using multifold in silico structure-based drug-designing approaches. A combination of molecular docking, pharmacokinetics examination, MD simulation, and MM/GBSA-based binding free energy calculations led us to discover six interesting chemical scaffolds that not only efficiently bind the target enzyme but also exhibit high binding free energy as compared to the reported substrate, suggesting their strong affinity for Isla. The binding of these molecules with Isla substantially affect the structural dynamics of Isla, which favors the inhibition of Isla function. Those identified new inhibitor candidates showed good drug-likeness, absorption, distribution, metabolism, and excretion profile in the comprehensive in silico analysis with no toxicity. These findings needs further in vitro and in vivo tests to scrutinize the effectivities of those noteworthy molecules as potential inhibitors of Isla to discover the new medications for Isla-related pathologies.

## Figures and Tables

**Figure 1 cancers-15-00901-f001:**
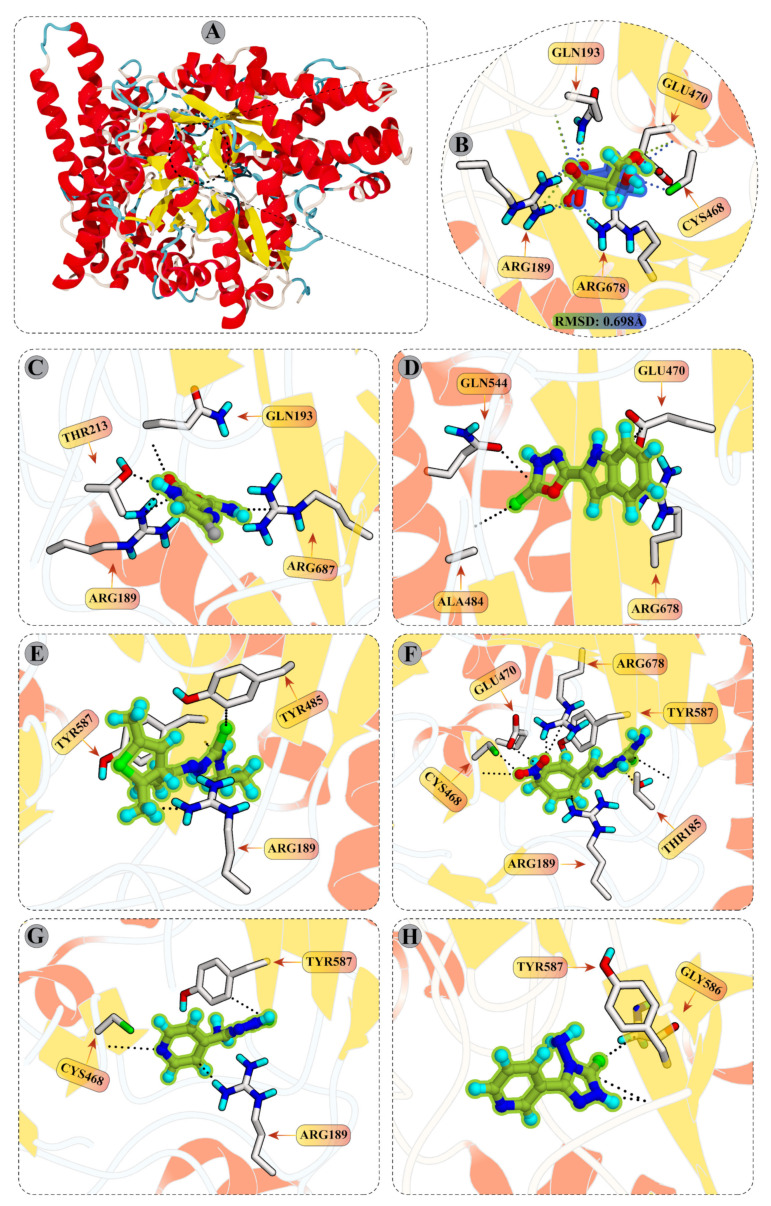
(**A**) The crystal structure of target enzyme, isethionate sulfite-lyase, is shown (PDB ID: 7KQ3); (**B**) the interactions of substrate (2-hydroxyethylsulfonic acid) with the active site residues are highlighted. The interactions of top selected hits from in-house database (**C**) CP1, (**D**) CP2, (**E**) CP4, (**F**) CP5, (**G**) CP8, and (**H**) CP9 with the active site residues are shown.

**Figure 2 cancers-15-00901-f002:**
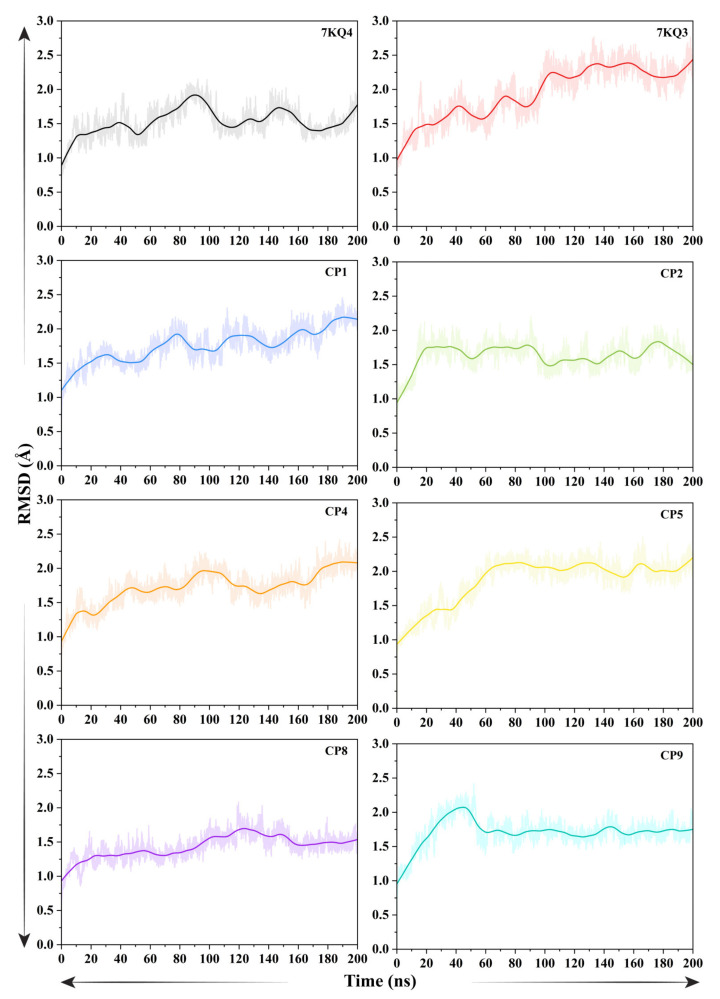
Root mean square deviation (RMSD) analysis of Isla in free state (7KQ4), reference-attached (7KQ3) and selected inhibitor candidates.

**Figure 3 cancers-15-00901-f003:**
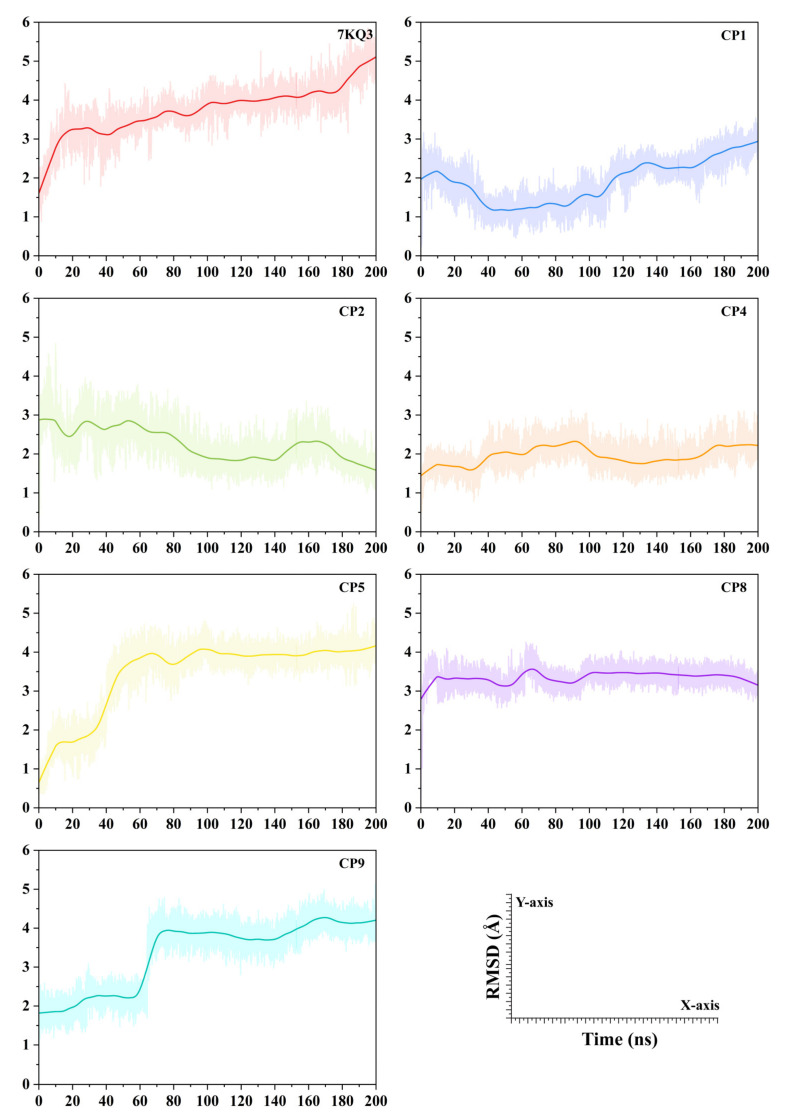
Ligands stability analyzed via root mean square deviation (RMSD) complex with Isla protein with reference attached system 7KQ4.

**Figure 4 cancers-15-00901-f004:**
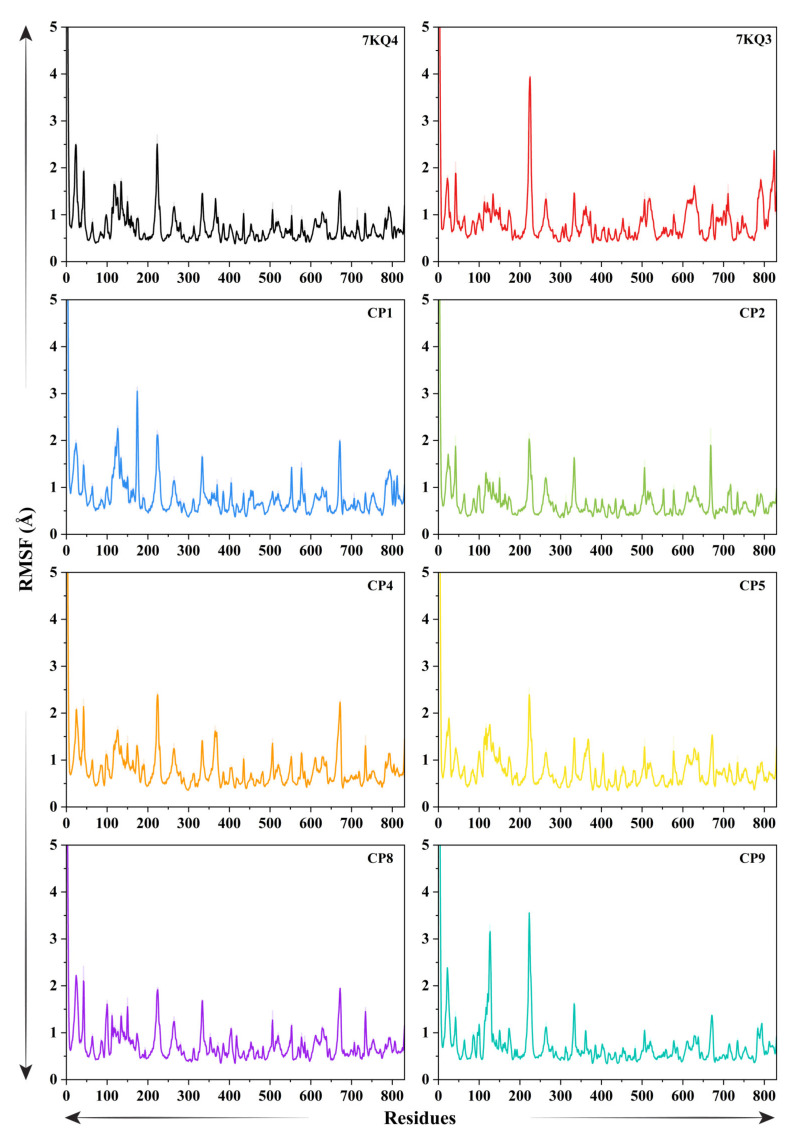
Graphical illustration of residual fluctuations of Isla via root mean square fluctuation (RMSF) in free state (7KQ4), reference complex (7KQ3), and selected ligands bound states. Each system is represented by a distinctive color.

**Figure 5 cancers-15-00901-f005:**
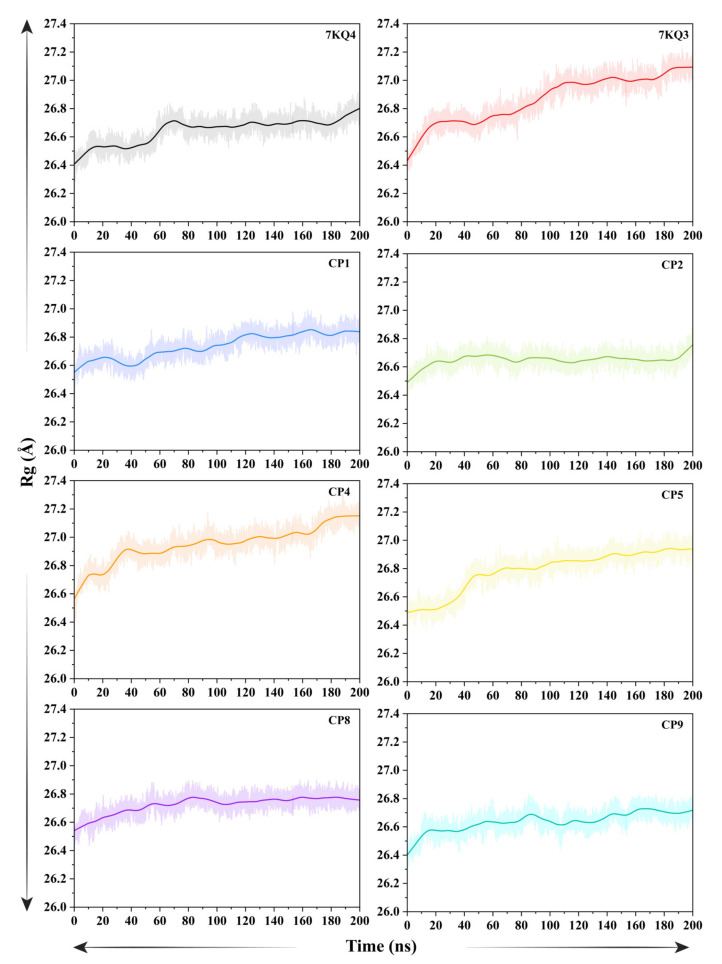
The radius of gyration (Rg) analysis plots of free stat (7KQ4), substrate-bound (7KQ3), and selected ligands-bound Isla.

**Figure 6 cancers-15-00901-f006:**
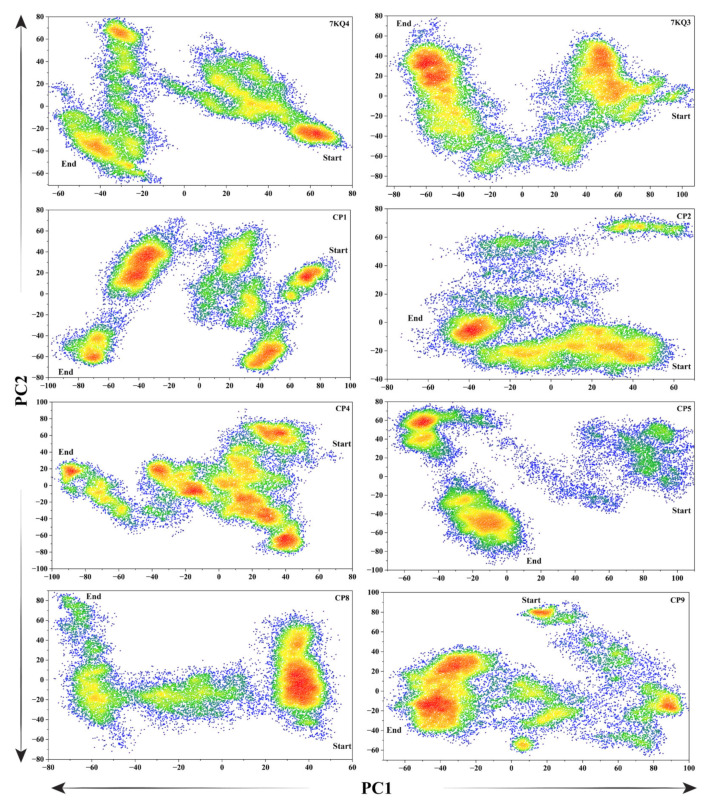
The principal component analysis of Isla in apo–Isla (7KQ4), substrate–Isla complex (7KQ3), and inhibitors bound states. The first principal component (PC1) and the second principal component (PC2) are plotted at x-axis and y-axis, respectively.

**Figure 7 cancers-15-00901-f007:**
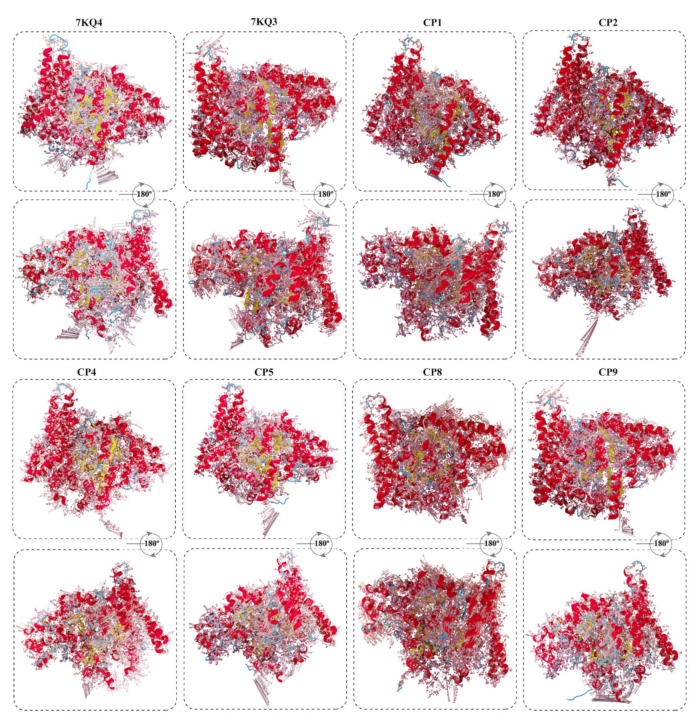
Principal motions of the Isla obtained from the principal component analysis (PCA) in free state (7KQ4), substrate-bound Isla (7KQ3), and inhibitors bound states. The length of the arrow represents the magnitude of the motion, and direction of the motion is denoted by the arrowhead.

**Figure 8 cancers-15-00901-f008:**
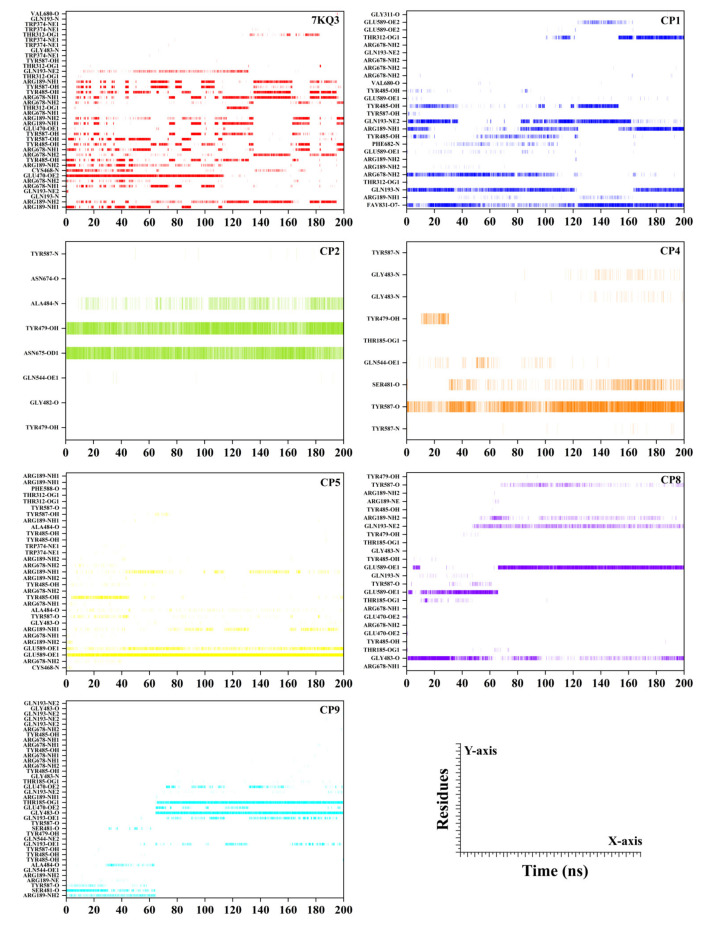
Hydrogen bond analysis of ligands with the active site residues of Isla during the simulation.

**Table 1 cancers-15-00901-t001:** The molecular interactions between compounds and the active site residues of isethionate sulfite-lyase.

Compound	Name	Docking Score	Ligand Atoms	Receptor Residues	Bonds	Distance (Å)	Energy (kcal/mol)
7KQ3	2-hydroxyethylsulfonic acid	−5.55	O6	OE2-GLU470	HBD	2.31	−0.9
			O4	NH1-ARG189	HBA	2.68	−7.0
			O4	N-GLN193	HBA	3.41	−0.8
			O5	NH2-ARG189	HBA	2.76	−7.2
			O5	NH2-ARG678	HBA	3.09	−5.5
			O65	N-CYS468	HBA	2.98	−1.9
			O76	NE2GLN193	HBA	2.94	−3.9
			O76	NH1-ARG678	HBA	3.16	−4.9
			O43	NH1-ARG189	Ionic	2.68	−7.0
			O43	NH2-ARG189	Ionic	3.43	−2.2
			O54	NH1-ARG189	Ionic	3.69	−1.2
			O54	NH2-ARG189	Ionic	2.76	−6.3
			O54	NH2-ARG678	Ionic	3.09	−3.9
			O76	NH1-ARG678	Ionic	3.16	−3.5
CP1	3-carbamoyl-5-fluoropyrazin-2-olate	−6.62	N6	OG1-THR312	HBD	2.88	−4.6
			N3	NH2-ARG678	HBA	2.89	−3.7
			O7	NH1-ARG189	HBA	2.76	−2.3
			O7	N-GLN193	HBA	3.06	−1.9
			O10	NH2-ARG189	HBA	2.76	−3.1
CP2	5-(1H-indol-2-yl)-1,3,4-oxadiazole-2(3H)-thione	−6.35	N7	OE2-GLU470	HBD	2.98	−2.2
			N13	OE1-GLN544	HBD	2.71	−6.6
			S15	N-ALA484	HBA	3.33	−1.9
			5-ring	NH2-ARG678	π-cation	3.43	−0.5
CP4	(E)-2-(1-(2,5-dimethylthiophen-3-yl)ethylidene)-N-ethylhydrazine-1-carbothioamide	−6.24	N14	O-TYR587	HBD	2.67	−1.8
			N3	NH2-ARG189	HBA	2.91	−1.1
			S13	CE1-TYR485	HBA	3.40	−0.5
CP5	(E)-2-(4-nitrobenzylidene)hydrazine-1-carbothioamide	−6.23	N4	OG1-THR185	HBD	2.95	−2.0
			N7	O-TYR587	HBD	2.92	−1.2
			N3	ANH2-RG189	HBA	2.87	−6.2
			S6	N-GLY483	HBA	3.20	−2.0
			S6	N-TYR587	HBA	3.23	−2.9
			O14	NH1-ARG678	HBA	2.78	−1.9
			O14	NH2-ARG678	HBA	3.18	−0.7
			O15	N-CYS468	HBA	2.82	−2.4
			O15	SG-CYS468	HBA	3.04	−0.8
CP8	4-amino-5-(pyridin-4-yl)-2,4-dihydro-3H-1,2,4-triazole-3-thione	−5.58	N2	N-CYS468	HBA	3.41	−2.5
			S13	N-TYR587	HBA	3.58	−0.9
			5-ring	NH2-ARG189	π-cation	3.12	−0.6
CP9	4-amino-5-(pyridin-3-yl)-2,4-dihydro-3H-1,2,4-triazole-3-thione	−5.35	N10	O-TYR587	HBD	2.93	−2.7
			S13	CA-GLY586	HBA	3.50	−1.6
			S13	N-TYR587	HBA	3.03	−3.4

HBA = hydrogen bond acceptor; HBD = hydrogen bond donor.

**Table 2 cancers-15-00901-t002:** Binding free energy calculations of selected compounds and substrate–Isla complexes.

ZINC ID	ΔE_vdW_ (kcal/mol)	ΔE_elec_ (kcal/mol)	ΔE_GB_ (kcal/mol)	ΔE_SURF_ (Å^2^)	ΔG_TOTAL_ (kcal/mol)
7KQ3	−10.52 ± 0.03	65.22 ± 0.18	−67.00 ± 0.14	−3.23 ± 0.01	−15.53 ± 0.08
CP1	−23.91 ± 0.03	−48.61 ± 0.07	48.90 ± 0.03	−3.23 ± 0.01	−26.85 ± 0.05
CP2	−30.88 ± 0.04	−35.00 ± 0.17	39.83 ± 0.16	−4.11 ± 0.01	−30.17 ± 0.04
CP4	−45.56 ± 0.03	−14.34 ± 0.01	31.14 ± 0.04	−5.50 ± 0.01	−34.27 ± 0.03
CP5	−36.43 ± 0.04	−57.58 ± 0.07	67.78 ± 0.06	−4.60 ± 0.01	−30.83 ± 0.04
CP8	−30.86 ± 0.03	−35.60 ± 0.04	48.71 ± 0.03	−3.79 ± 0.01	−21.54 ± 0.04
CP9	−31.46 ± 0.03	−36.74 ± 0.06	48.98 ± 0.04	−4.00 ± 0.01	−23.22 ± 0.04

## Data Availability

The data can be shared up on request.

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
