# Peer review of "Multi-Fold Computational Analysis to Discover Novel Putative Inhibitors of Isethionate Sulfite-Lyase (Isla) from *Bilophila wadsworthia:* Combating Colorectal Cancer and Inflammatory Bowel Diseases"

_cancers, 2023, doi:10.3390/cancers15030901_

Round 1

Reviewer 1 Report

Thank you for the opportunity to review the article entitled: Multi-Fold Computational Analysis to Discover Novel Inhibi-2 tors of Isethionate Sulfite-Lyase (Isla) From Bilophila 3 Wadsworthia: Combating Colorectal Cancer And Inflammatory 4 Bowel Diseases by Waqas et al. The article is very interesting and the methods are conducted in a manner appropriate for the research goal. However, there are several things that ought to be considered before publication.

1.    Throughout the main text the authors  refer the compounds that are being in silico examined as “inhibitors”. There is no evidence presented in this article that these compounds have inhibitory effects on IslA. It should be rephrased throughout the text that these are “putative” inhibitors, or “candidate” inhibitors, or “potential” inhibitors.

2.    In a similar vain as the comment above, this is a “discovery” paper. In line 28 you describe this as “structure-based drug discovery”. This is accurate. However, there are places in the text where the authors try to descrdibe their methods as a “design” approach to drug discovery (e.g., line 27; “Here we describe the structure-based designing”). “Designing is not accurate.

Title: IslA not Isla

Line 24: Incomplete sentence

Line 25 and throughout: Subscript the 2 in H2S

Line 56-58: Need to reference Peck et al. 2019 PNAS 116(8)3171: A glycyl radical enzyme enables hydrogen sulfide production by the human intestinal bacterium Bilophila Wadsworthia

Line 88: This paper does NOT identify small inhibitors: No data is shown of enzyme function inhibition. Throughout the text these should be referred to as "putative" or "potential"

Line 226-231: needs to reference Dawson paper

Line 228: Figure 1a

Line 232: there is not design in this manuscript

Figure 9: increase image resolution

Supplementary Tables need more in-depth descriptions

Author Response

Thank you for the opportunity to review the article entitled: Multi-Fold Computational Analysis to Discover Novel Inhibitors of Isethionate Sulfite-Lyase (Isla) From Bilophila Wadsworthia: Combating Colorectal Cancer And Inflammatory Bowel Diseases by Waqas et al. The article is very interesting, and the methods are conducted in a manner appropriate for the research goal. However, there are several things that ought to be considered before publication.

1.Throughout the main text the authors  refer the compounds that are being in silico examined as “inhibitors”. There is no evidence presented in this article that these compounds have inhibitory effects on IslA. It should be rephrased throughout the text that these are “putative” inhibitors, or “candidate” inhibitors, or “potential” inhibitors.

Response: Thank you for the fruitful comments. As per suggestion we have referred our identified compounds as “putative” inhibitors, or “candidate” inhibitors, or “potential” inhibitors or ligands in the text. All the changes are highlighted in yellow color in the manuscript.

2.In a similar vain as the comment above, this is a “discovery” paper. In line 28 you describe this as “structure-based drug discovery”. This is accurate. However, there are places in the text where the authors try to describe their methods as a “design” approach to drug discovery (e.g., line 27; “Here we describe the structure-based designing”). “Designing is not accurate.

Response: We have made the corrections according to the reviewer’s suggestion

Title: IslA not Isla

Response: Corrections have been made throughout the manuscript

Line 24: Incomplete sentence

Response: Corrected

Line 25 and throughout: Subscript the 2 in H2S

Response: Corrected

Line 56-58: Need to reference Peck et al. 2019 PNAS 116(8)3171: A glycyl radical enzyme enables hydrogen sulfide production by the human intestinal bacterium Bilophila Wadsworthia

Response: Cited

Line 88: This paper does NOT identify small inhibitors: No data is shown of enzyme function inhibition. Throughout the text these should be referred to as "putative" or "potential"

Response: Corrected

Line 226-231: needs to reference Dawson paper

Response: Dawson paper is cited at reference # 20.

Line 228: Figure 1a

Line 232: there is not design in this manuscript

Response: Corrected

Figure 9: increase image resolution.

Response: The image resolution was increased for clear visibility.

Previous figure 9, now figure 7 is presented in high resolution

Supplementary Tables need more in-depth descriptions

Response: Thank you for the rectification. The descriptions are now added in tables in supporting information and all the mistakes are now rectified.

Reviewer 2 Report

The manuscript entitled “Multi-fold computational analysis to discover novel inhibitors of isethionate sulfite-lyase (Isla) from Bilophila Wadsworthia: combating colorectal cancer and inflammatory bowel diseases” by Waqas et.al reported screening of inhibitors targeting a glycyl radical enzyme, IslA, important in providing the terminal electron acceptor for anaerobic respiration. The authors claim that inactivating this enzyme can prevent the disease-linked bacterium from releasing H2S that aids in development of colon cancer. The authors employed different computational methods including molecular docking, pharmacokinetics properties evaluation, binding free energy calculations and MD simulations to screen and assess potential inhibitors. Out of a small library of 800 chemical compounds, they identified and analyzed 9 potential inhibitors. The choice of the target is interesting and perhaps the work is the first piece towards inhibiting this enzyme family that has many unique properties for drug development purpose. However, the reviewer thinks that there are many issues of the manuscript in its current version.

1.       The authors are keen for clearing H2S in human gut to prevent/treat colon cancers. Yet they didn’t discuss in depth the connection of H2S to colon cancers and cite appropriate references. e.g the amount of H2S, the stage of colon cancer and the molecular basis.

2.       All methods used by the authors are in silico. With the drug development purpose that the authors claim, the number of compounds in their in-house database seems to be way too small.  In this case biochemical evidence to support their computational findings becomes vital but is nevertheless missing. Glycyl radical enzyme is difficult to work with. Formation of its glycyl radical cofactor essential for the enzyme activity requires an Fe-S cluster containing activating enzyme belonging to the radical SAM enzyme family. Reconstituting these oxygen sensitive cofactors is technically challenging and requires special expertise. How would the compounds identified be verified as truly potent inhibitors? At least I would suggest a simple experiment to test them. Bilophila wardsworthia can be cultured in the defined medium with taurine as the sole carbon source and electron acceptor. Addition of the inhibitors of IslA is expected to inhibit the bacterial growth.

3.        In the human gut, many other food-derived sulfonates can serve as the respiration terminal electron acceptor for Bilophila wadsworthia. How would the authors envision solely inhibiting IslA be sufficient to prevent formation of H2S? At least the authors should discuss the potential pitfall of their strategy. The authors should note that another glycyl radical enzyme HpsG would similarly break the C-S bond of dihydroxypropanesulfonate (Liu et al PNAS 117 (27) 15599-15608). Bilophila wadswarthia has been reported capable of using a variety of sulfonates including sulfoacetate, cysteate and sulfolactate etc. as respiration electron acceptor with all producing H2S. The authors should perform thorough literature search and know better of the physiology of this disease linked bacterium.

4.       Despite many computational methods used in this study, the authors seem to focus only on the biophysical properties of the enzyme and miss the high reactivity of its radical cofactor. Would it be quenched? Would there be inhibitor radical formed? How would the chemistry affect binding of the inhibitor?

5.       The manuscript is missing important references. The first report on IslA was not even cited  (Proc Natl Acad Sci U S A. 2019 Feb 19;116(8):3171-3176). The first structure of IslA also named IseG (PDB code 5YMR) reported in Nat. Comm. 2019 Apr 8;10(1):1609 was not even cited. Given that other sulfate and sulfite reducing bacteria and even fermenting bacteria contain IslA for generation of respiration electron acceptor (Proc Natl Acad Sci U S A. 2019 Feb 19;116(8):3171-3176, Nat. Comm. 2019 Apr 8;10(1):1609) and for sulfur assimilation respectively (Biochem J. 2019 Aug 15;476(15):2271-2279), how would IslA inhibitors impact the gut microbiome, colon cancer and human health?

6.       The authors should check that the significant figures used in data reporting are appropriate.

Author Response

The manuscript entitled “Multi-fold computational analysis to discover novel inhibitors of isethionate sulfite-lyase (Isla) from Bilophila Wadsworthia: combating colorectal cancer and inflammatory bowel diseases” by Waqas et.al reported screening of inhibitors targeting a glycyl radical enzyme, IslA, important in providing the terminal electron acceptor for anaerobic respiration. The authors claim that inactivating this enzyme can prevent the disease-linked bacterium from releasing H2S that aids in development of colon cancer. The authors employed different computational methods including molecular docking, pharmacokinetics properties evaluation, binding free energy calculations and MD simulations to screen and assess potential inhibitors. Out of a small library of 800 chemical compounds, they identified and analyzed 9 potential inhibitors. The choice of the target is interesting and perhaps the work is the first piece towards inhibiting this enzyme family that has many unique properties for drug development purpose. However, the reviewer thinks that there are many issues of the manuscript in its current version.

1.The authors are keen for clearing H2S in human gut to prevent/treat colon cancers. Yet they didn’t discuss in depth the connection of H2S to colon cancers and cite appropriate references. e.g the amount of H2S, the stage of colon cancer and the molecular basis.

Response: The connection of H2S production with colon cancer is now discussed in detail and several references are now cited in the text.

  1. All methods used by the authors are in silico. With the drug development purpose that the authors claim, the number of compounds in their in-house database seems to be way too small.  In this case biochemical evidence to support their computational findings becomes vital but is nevertheless missing. Glycyl radical enzyme is difficult to work with. Formation of its glycyl radical cofactor essential for the enzyme activity requires an Fe-S cluster containing activating enzyme belonging to the radical SAM enzyme family. Reconstituting these oxygen sensitive cofactors is technically challenging and requires special expertise. How would the compounds identified be verified as truly potent inhibitors? At least I would suggest a simple experiment to test them. Bilophila wardsworthiacan be cultured in the defined medium with taurine as the sole carbon source and electron acceptor. Addition of the inhibitors of IslA is expected to inhibit the bacterial growth.

Response: Thank you for the fruitful suggestion. Due to unavailability of in-vitro enzyme inhibition facilities of this enzyme in our lab, we are unable to test the potency of the identified hits at this moment. However, we will establish the assay and test these compounds in future. In addition, we would be happy to collaborate with scientific community for testing of our compounds, that’s the reason we focused to publish our current in-silico hypothesis.

  1. In the human gut, many other food-derived sulfonates can serve as the respiration terminal electron acceptor for Bilophila wadsworthia. How would the authors envision solely inhibiting IslA be sufficient to prevent formation of H2S? At least the authors should discuss the potential pitfall of their strategy. The authors should note that another glycyl radical enzyme HpsG would similarly break the C-S bond of dihydroxypropanesulfonate (Liu et al PNAS 117 (27) 15599-15608). Bilophila wadswarthiahas been reported capable of using a variety of sulfonates including sulfoacetate, cysteate and sulfolactate etc. as respiration electron acceptor with all producing H2S. The authors should perform thorough literature search and know better of the physiology of this disease linked bacterium.

Response: Several food-derived sulphonates are present in the human gut, but we focus on targeting Bilophila wadsworthia Isla because it was in high concentration in samples taken from cancer patients. Moreover, until now, no inhibitor or drug molecule has been reported for this enzyme. This is the novelty of this study.

Another glycyl radical enzyme HpsG (Liu et al PNAS 117 (27) 15599-15608) will be targeted in future and published separately.

Further physiology of cancer linked with Bilophila wadsworthia is now discussed in detail in the revised manuscript.

  1. Despite many computational methods used in this study, the authors seem to focus only on the biophysical properties of the enzyme and miss the high reactivity of its radical cofactor. Would it be quenched? Would there be inhibitor radical formed? How would the chemistry affect binding of the inhibitor?

Response: we highlighted the importance of Isla inhibition because of its novelty and applying computational resources for the identification of Isla inhibitors. However, the reaction mechanism analysis is not the main scope of this manuscript.

The sulfate moiety in the substrate is removed during the reaction and produces the H2S, the radical formed. In our selected molecules, sulfate moiety is not present like the substrate of the enzyme. Therefore, we did not hypothesize the removal of H2S and sulfate. 

  1. The manuscript is missing important references. The first report on Isla was not even cited (Proc Natl Acad Sci U S A. 2019 Feb 19;116(8):3171-3176). The first structure of IslA also named IseG (PDB code 5YMR)reported in Nat. Comm. 2019 Apr 8;10(1):1609 was not even cited. Given that other sulfate and sulfite reducing bacteria and even fermenting bacteria contain IslA for generation of respiration electron acceptor (Proc Natl Acad Sci U S A. 2019 Feb 19;116(8):3171-3176, Nat. Comm. 2019 Apr 8;10(1):1609) and for sulfur assimilation respectively (Biochem J. 2019 Aug 15;476(15):2271-2279), how would IslA inhibitors impact the gut microbiome, colon cancer and human health?

Response: we have particularly focused on identifying drug-like molecules of Isla from Bilophila wadsworthia because these bacteria were isolated from cancer patients samples.

The first report on IslA (Proc Natl Acad Sci U S A. 2019 Feb 19;116(8):3171-3176) is now cited in the text (Ref-21).

The first structure of IslA also named IseG (PDB code 5YMR) reported in Nat. Comm. 2019 Apr 8;10(1):1609 is now cited in the text (Ref-26).

Th significance of IslA inhibitors and their impact on gut microbiome, colon cancer and human health is now discussed in the manuscript.

  1. The authors should check that the significant figures used in data reporting are appropriate.

Response: Two figures are now moved in supporting information and only significant figures are used in the main text.